# Computing Tractable Probabilistic Models: A Hardware Perspective

**Martin Andraud**[1,3]  **Lingyun Yao**[1]  **Karthekeyan Periasamy**[1]  **Jelin Leslin**[1]  **Martin Trapp**[2]

[1]Dept. of Electronics and Nanoengineering, Aalto University, Finland
[2]Dept. of Computer Science, Aalto University, Finland
[3]ICTEAM, UCLouvain, Belgium

## Abstract

Several deep learning models recently raised overconfidence and reliability concerns, and alternative models for trustworthy and explicit decision-making systems are on the rise. Among them, tractable probabilistic models (TPMs) have recently gained significant interest, exploiting their tractability for energy-efficient and general-purpose inference. Yet, although "software" implementations of TPMs have shown great potential, the hardware computation and *acceleration* of these models is still largely underexplored. In this work, we offer a perspective on why this is the case, and elaborate on what can be done to design more efficient processors suited for TPMs. Our analysis shows that although research seems currently fragmented, several pieces of the puzzle can be combined to enable a larger use and a more efficient computation of TPMs in edge AI systems.

## 1 INTRODUCTION

The development of explainable, compact and trustworthy AI is essential for the next generation of decision-making systems, targeting applications in healthcare, smart systems, or automotive [Ghahramani, 2015]. In this context, deep neural networks (NNs) have become a *de facto* standard, providing at the same time state-of-the-art performance and possibilities of efficient hardware computation [Hooker, 2020]. Yet, DNNs have many known limitations, whether considering their low interpretable nature or overconfidence [Nalisnick et al., 2019], making them potentially unsuited for explainable and trustworthy AI. Moreover, the drastic increase in DNN model size and inference costs [Marcus, 2020, Heljakka et al., 2023] introduces challenges in their computation on resource-constrained hardware. Thus, alternative or complementary models are explored, such as

probabilistic models (PMs) [Ghahramani, 2015], allowing for explicit probabilistic inference. A particularly promising direction in this line of work is exploiting tractable probabilistic models (TPMs), as they enable provably exact and efficient inference in many scenarios and have shown to be successful in many settings including speech or image recognition [Nicolson and Paliwal, 2020, Wang and Wang, 2018, Stelzner et al., 2019], semantic mapping [Zheng and Pronobis, 2019], outlier detection [Peharz et al., 2020] or neurosymbolic AI, including uncertainty estimation [Kang et al., 2024] or rule-based integration [Maene et al., 2025].

Naturally, many of those applications will benefit from being *embedded*, *i.e.* computed online by dedicated *edge devices*. Yet, the development of hardware *accelerators* for TPMs is still in its infancy compared to their DNN counterparts, practically limiting their use in resource-constrained applications (see Fig. 1 (a)). In this work, we aim to provide a perspective on this challenge by answering three questions: (i) What makes TPMs more difficult to compute on hardware (Section 2)? (i) What are the current trends in hardware acceleration of TPMs (Section 3)? (iii) What still needs to be done for the next generation of TPM accelerators (Section 4)?

## 2 COMPUTING TPMs

In this section, we target the first of our questions: *What makes TPMs more difficult to compute on hardware?*

### 2.1 PROBABILISTIC CIRCUITS AS TPMs

To discuss challenges related to TPMs, we use the framework of probabilistic circuits (PCs) [Choi, 2022]. PCs are probabilistic models that use computational graphs, comprising arithmetic operations, such as weighted sums $\oplus$ and multiplication $\otimes$. In PCs, many inference queries can be answered with a single pass through the circuit, making them particularly suited for embedded scenarios. To

*Accepted for the 8th Workshop on Tractable Probabilistic Modeling at UAI  (TPM 2025).*

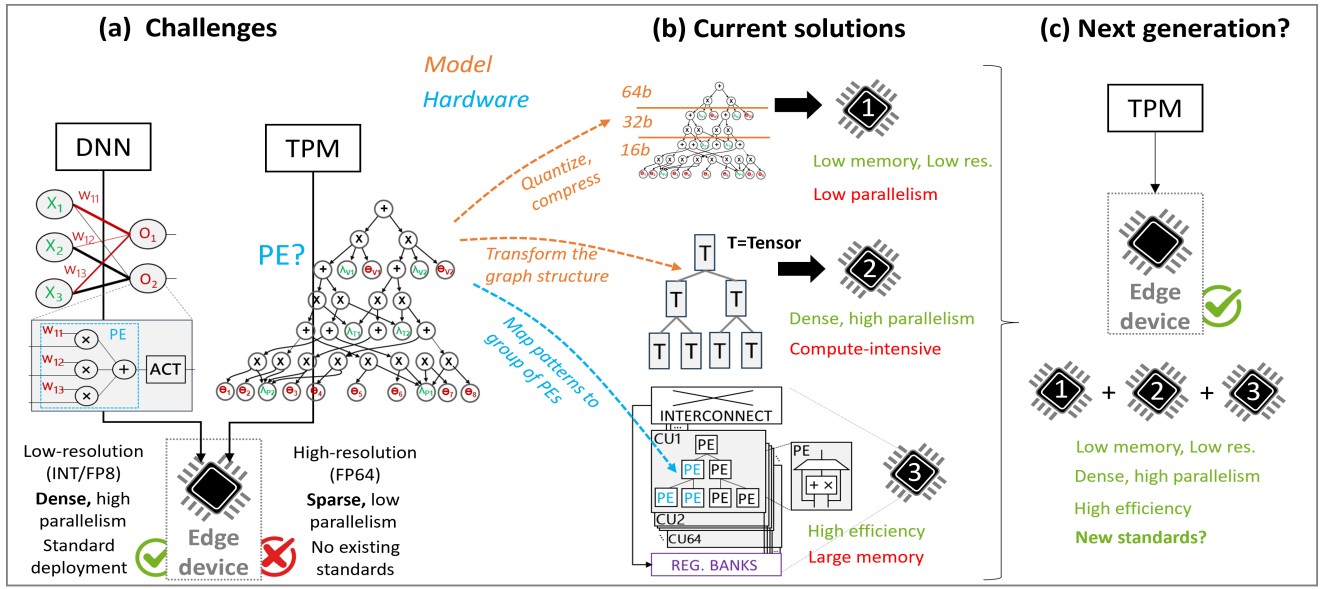

Figure 1: Illustration of the current challenges, proposed solutions and opportunities for the next generation TPM accelerators.

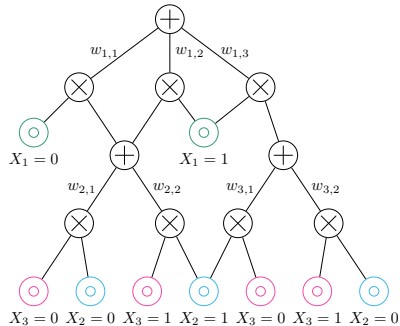

Figure 2: Example of a probabilistic circuit.

make the analogy with deep NNs, PCs can be decomposed into two operations, multiply-and-accumulate and product operations. Sum nodes are computed as a weighted sum, corresponding to multiply-and-accumulate, and product nodes correspond to product operations. Fig. 2 illustrates a PC over discrete variables ($X_1, X_2, X_3$), representing the joint distribution $P(X_1, X_2, X_3)$ through a computational graph consisting of weighted summations and multiplications[1]. PCs have recently been vectorised for better scalability and trained in mainstream deep learning frameworks such as Tensorflow and PyTorch (*e.g.*, [Peharz et al., 2020, Loconte et al., 2024, Liu et al., 2024]).

In view of their increased use in various applications, the hardware acceleration of PCs has gained interest, targeting a whole range of platforms, such as CPUs/GPUs [Sommer et al., 2021, Liu et al., 2024], FPGAs [Sommer et al., 2018, 2020, Kruppe et al., 2022, Choi et al., 2022, Periasamy et al.,

---

[1]The reader is referred to [Choi, 2022] for an in-depth introduction to PCs.

2024, Zhang et al., 2025] and custom Application Specific Integrated Circuits (ASICs) [Shah et al., 2021a, 2022].

## 2.2 SPECIFICITY OF PC ACCELERATION

Generally, designing a good AI accelerator requires *fast* and *efficient* hardware, and is done in three steps: (1) identifying the most hardware-costly operations and building dedicated Processing Elements (PEs) to execute them efficiently, (2) reusing these PEs as much as possible for maximum efficiency, and (3) executing the PEs in parallel to maximise the hardware speed (throughput). As such, accelerators are benchmarked regarding their speed (in operations per second, OPS) and their efficiency (in OPS/W). As shown in Fig. 1 (a), these principles apply well to deep NNs, because: (1) the most intensive operation is multiply-and-accumulate (MAC), which can be done very efficiently at low resolution on a dedicated PE, (2) MAC is performed by every neuron of the model, leading to a large PE reuse, and (3) many neurons (hence PEs) can be computed in parallel to increase the throughput. In addition, there exist common development frameworks to directly port DNNs into hardware.

Yet, this does not apply to PCs, which are generally high-resolution and more irregular, complicating these hardware acceleration principles. We consider the following two challenges:

**Graph Irregularity**. Typical 'scalar' PCs are relatively sparse, especially compared to deep NNs. This sparsity reduced the performance of classical accelerators. Hence, as analyzed in [Sommer et al., 2020], computing a PC on highly parallel GPUs leads to poor performance, as it is difficult to identify repeated computations and parallel threads.

**Computation resolution.** As variables to be computed represent probabilities, they lie in the range $[0, 1]$. These values are successively added and multiplied, leading to potentially extremely small probabilities at the top layers. Thus, computing PCs require a high resolution (typically $30 - 40$ floating point bits for medium-sized PCs [Shah et al., 2019, Sommer et al., 2020]), requiring double precision.

## 3 EXISTING TPM ACCELERATORS

Based on these aforementioned challenges, we discuss currently implemented solutions, as depicted in Fig. 1 (b): *What are the current trends in hardware acceleration of TPMs?*

Let us consider "software" PCs as a baseline, *i.e.*, effectively computed in a generic processor (CPU or GPU). Generic platforms use a linear 32-bit or 64-bit floating point representation, hence computing PCs (especially during training) can quickly lead to *underflow*. An underflow occurs when a computed probability falls below the smallest representable number due to limited precision, causing it to be treated as zero, which is detrimental to performance and learning. In this case, logarithmic computation is used, where multiplications become additions and additions are efficiently computed using the "Log-Sum-Exp" (LSE) trick,

$$\text{LSE}(x_1, x_2, \ldots, x_n) = c + \log \left( \sum_{i=1}^{n} e^{x_i - c} \right), \quad (1)$$

where $c = \max(x_1, x_2, \ldots, x_n)$ and $x_i$ represent log probabilities. Note that the LSE function introduces a constant $c$, which scales the exponential values to prevent underflow. Smaller values are shifted before the linear transfer, effectively preventing underflow when exponenting the log probabilities. Here, log values $x_i$ are encoded in 32-bit or 64-bit floating point, resulting in a large effective range.

On top of LSE, there has been interest in compiling and accelerating inference on CPUs/GPUs, for instance, using the SPNC framework [Sommer et al., 2021]. SPNC uses a workflow to compile the PC for CPU or GPU execution, based on specific compiler architectures. $500\times$ to $800\times$ gains are observed compared to baseline learning algorithms (SPFlow), showing that there exist large optimization possibilities.

### 3.1 PCs ON DEDICATED HARDWARE

Compared to utilizing generic processors, considerably larger speed-up and efficiency can be found with dedicated hardware, spanning across FPGAs (*i.e.*, reconfigurable hardware) and ASICs (*i.e.*, dedicated chips). Note that a direct comparison between different hardware is challenging, as the computation format, type of PC and benchmarks vary. Hence, we abstract this comparison into three aspects:

**Quantize and compress?** From the existing literature, it is clear that not all nodes in a PC need the same computation resolution, enabling the reduction of the hardware cost [Periasamy et al., 2024, Sommer et al., 2020]. Hence, quantization and compression techniques are a promising avenue. For instance, pruning and growing can change the shape of the PC to learn better models [Dang et al., 2022]. Going further, recent works have shown that PC compression is possible [Zhang et al., 2025] and can reduce hardware and memory costs by around 50% on average, compared to the initial PC model. As a drawback, although limiting memory and hardware use, these methods do not solve the (lack of) parallelism of PCs, that remains an issue.

**How to identify computation patterns?** The optimal structure of a PC inherently depends on the encoded probability distribution. Thus, it is challenging to find one (or a few) repeated computations to form PEs. We consider two approaches to tackle this issue: (1) change the model's structure, or (2) adapt the hardware architecture. Regarding (1), tensorized PCs, such as RAT-SPN [Peharz et al., 2019] or Einsum networks [Peharz et al., 2020] allow to fix a computing structure and replicate it over the graph, significantly increasing the throughput on GPU devices during training and inference. Regarding (2), accelerators are generally coupled with a dedicated compiler to *group* computation 'patterns' and accelerate them in dedicated blocks (if possible in parallel). One example of such a compiler tool is GraphOpt [Shah et al., 2021a], coupled with the DPU_V2 accelerator [Shah et al., 2022]. Here, each pattern is mapped to a given PE *tree*, increasing the accelerator's performance. Although offering more parallelism to compute the model, transforming the structure can lead to models that are more compute-intensive than scalar PCs.

**Linear or Log computing?** Generally, hardware accelerators typically prefer linear computing systems. This is motivated by two reasons: (1) 'exact' linear hardware is generally much more efficient than logarithmic [Sommer et al., 2018], and (2) 'approximate' log computation, while more efficient, introduces errors that are not suitable for high-resolution PC computation. In addition, implementing direct LSE necessitates customized hardware functions, because it still contains explicit $log()$ and $exp()$ operations, even using already existing logarithmic computing hardware blocks. Those functions can still be relatively costly to implement, thus hardware implementations are limited. Hence, most accelerators rely on floating-point (FP) [Sommer et al., 2020, Zhang et al., 2025] or Posit [Shah et al., 2021b] number systems, while other FPGA platforms allow the user to choose [Sommer et al., 2018, Periasamy et al., 2024]. Regarding (1), using a custom resolution (*i.e.* not only single or double precision) is desirable for efficiency. This can be done by theoretically determining the best resolution [Shah et al., 2019], by iterative search [Sommer et al., 2020] or a mix between them [Periasamy et al., 2024]. The PC can also

be effectively quantized to further reduce its memory footprint [Zhang et al., 2025]. Regarding point (2), approximate computing has been recently studied to replace exact multiplications, obtaining significant power savings on FPGA hardware while limiting the accuracy loss [Yao et al., 2024]. While this direction is promising, further research is needed.

**Model-generic or Model-specific accelerator?** Model-specific accelerators directly replicate the PC graph into elementary computation blocks and compute them sequentially. This provides a tailored computation for each PC graph, yet requiring reconfigurable hardware (FPGA) [Sommer et al., 2018, 2020, Kruppe et al., 2022, Periasamy et al., 2024, Zhang et al., 2025]. In contrast, *model-generic* accelerators use a more generic approach, necessary when developing ASICs. The hardware typically comprises parallel PEs, each computing a small part of the PC. Hence, any PC can be fitted using a dedicated compiler/scheduler. The first example is DPU [Sommer et al., 2018], comprising 64 parallel PEs, each PE implementing an addition or a multiplication. A main feature of the design is the use of load/store streaming units and a local scratchpad, enhancing data reuse and allowing $12\times$ gains compared to a GPU. This architecture has been improved, based on the GraphOpt compiler [Shah et al., 2021a], leading to DPU-v2 [Shah et al., 2022], containing parallel trees of PEs, each able to perform an addition, a multiplication or being bypassed. However, the final performance gains are limited by inherent trade-offs in hardware design (essentially, adding more flexibility requires more hardware, in turn degrading efficiency and/or throughput).

**The current state of hardware acceleration.** Based on the 'map' of existing PC-specific hardware shown in Fig. 3, we can see that while ASICs are $10-100\times$ more efficient than FPGA implementations (one order of magnitude efficiency corresponds to the space between two diagonal lines). For instance, DPU-v1 shows a throughput of 33.7 Giga Operations per second (GOPS) and a peak energy efficiency of 248 GOPS/W, which is around $10\times$ better than FPGA-based accelerators. DPU-v2 improved the speed to be 34.6 GOPS, with a peak energy efficiency of 31.6 GOPS/W. However, as it can be seen, their computing speed stays in the same order of magnitude, which seems to indicate that the current models they accelerate can't be further parallelized. In the next section, we provide directions to improve the current situation.

# 4 TRENDS AND OPPORTUNITIES

We arrive to the last question: *What still needs to be done for the next generation of TPM accelerators?*. The previous literature study offers two opportunities (O):

(O1): Even though the computation resolution remains an issue, smart *quantization* and *compression* techniques are

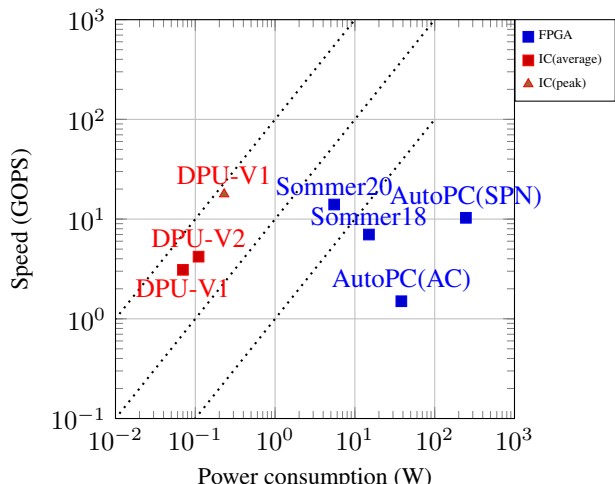

Figure 3: Comparison of academic accelerators for PCs: "Sommer18" is [Sommer et al., 2018] "Sommer20" is [Sommer et al., 2020], "AutoPC" is [Periasamy et al., 2024], "DPU-V1" is [Shah et al., 2021a], "DPU-V2" is [Shah et al., 2022]

developed but remain to be applied at larger scales. For model-generic ASICs, reconfiguration is key, although it is yet to be combined with online mixed-precision computation. Alternatively, efficient log-computing frameworks are crucial to enable accurate and efficient PC computation.

(O2): The irregularity challenge is mostly tackled by tighter hardware/software integration. At the model level, this can be enabled through vectorization, with proven results on GPUs and pushed by recent tensor factorization software such as CirKit [Lab, 2024] that are yet to be demonstrated with dedicated hardware implementations. Regarding the hardware itself, novel compilation frameworks and processor architectures are both necessary to improve the throughput of current accelerator and reduce the memory overheads.

# 5 CONCLUSION AND PERSPECTIVES

There is now a large body of literature tackling the acceleration of TPMs are various abstraction levels. We expect to see significant gains in accelerator's performance when combining the different solutions, ideally through new standards in terms of model representation and compilation (see Fig. 1 (c)). When becoming a reality, TPMs could become the new winners of the hardware lottery [Hooker, 2020], alongside deep NNs.

# 6 ACKNOWLEDGEMENTS

This work has been funded by two Research Council of Finland's projects (grants 332218 (MA) and 347279 (MT)) and the EU Pathfinder project SUSTAIN.

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
