# OpenReview forum: "Computing Tractable Probabilistic Models: A Hardware Perspective"
_auai.org/UAI/2025/Workshop/TPM — TPM 2025_

### Official Review · Reviewer_U2mS · 2025-06-15
**Interesting and relevant, but who (which community) is it for?**

**Rating:** 3

**Review:**

The paper presents several current challenges in realizing probabilistic circuits for hardware.

I do not have a background in hardware, so I am not very confident of my evaluation.

I believe the paper is of interest to the TPM community, however the presentation seems to be
directed towards the hardware community. I believe a more accessible text directed to the TPM
community would be of more value.

Below are some pointwise comments.

Typo in the second paragraph of Section 1: (i) -> (ii).

Second paragraph of Section 2.1: "the hardware acceleration of PCs has gained interest": I believe
this should be something like, there has been a gain in interest for hardware acceleration, not
that hardware acceleration has gained an interest in something.

Section 2.2, Graph Irregularity: "This sparsity reduced" -> "This sparsity reduces"

In the abstract: double quotes are used and in Section 2.2 Graph Irregularity, single quotes are
used. One should use one or the other.

Section 2.2, Computation resolution: (0, 1) -> [0, 1].

Section 2.2, Computation resolution: I am not familiar with hardware, but both in NNs and PCs,
values are usually computed in logspace. This is referenced in the following section as being
costly to compute in hardware, but some discussion should be presented in this paragraph.

Typo: "log probabilitites" -> "log probabilities" after Eq 1.

Section 3.1, Quantise and compress?: "Although limiting memory and hardware use, these methods do
not solve..." -> "Although these methods limit memory and hardware use, they do not solve..."

Section 3.1, How to identify computation patterns?: RAT-SPN and EinsumNets are cited as *recent*
works, however these are established papers from 5-6 years ago.

Yao et al citation is missing the year.

Section 3.1, Model-generic or model-specific accelerator?: The acronym IC is used without
defining it. I assume it stands for integrated circuit? Given the TPM community is not familiar
with hardware, I would recommend defining it. Also "in turns degrading" -> "in turn degrading".

Section 3.1, The current state of hardware acceleration: Again, since the TPM community is not
familiar with hardware, I would recommend explaining what GOPS stands for (is it giga operations
per second?) and what 0.9V and 8b represent.

There are duplicate references, e.g. Sommer et al 2018 and Sommer et al 2020 both are duplicated.

I believe vectorization (of circuits) is being used as a synonym of tensorization (of circuits),
but both terms are used concurrently. I would defer to using tensorization only as that is the more
common term.

Typo in Section 5: "TPMs are various" -> "TPMs at various"

---

### Official Review · Reviewer_n5Gv · 2025-06-15
**The paper discusses key challenges and opportunities in accelerating probabilistic circuits**

**Rating:** 2

**Review:**

This paper discusses key challenges and opportunities in accelerating Probabilistic Circuits (PCs). The paper discusses two aspects that are very different from accelerating neural networks: (i) the irregular and sparse structure of PCs, and (ii) the need for high-precision calculations (or log-space calculations).

The paper further discusses some current techniques to accelerate PCs, including quantization/compression, the development of custom hardware processing elements, and more parallelizable (or dense) PC structures.

The paper provides a nice overview of the systems-side developments of PCs. However, I found the elaboration to be a bit scattered. For example, the paper mentioned dense and sparse PCs. But it is not very clear which techniques/systems should be used in each case. I think the paper would benefit from some rearrangements of the text to separate the challenges induced by PCs and the solutions provided by existing methods. This could provide a better overview on which solutions are more effective in solving a certain challenge.